# Experimental Evaluation of Multipath Mitigation in TDOA-Based Indoor Passive Localization System Using A Beam Steering Broadband Circular Polarization Antenna

**Changjiang Su [1]**, **Yanqun Liu [1]**, **Leilei Liu [2]**, **Mei Yang [1,3]**, **Hongxin Zhao [1]** and **Xiaoxing Yin [1,*]**

[1]  State Key Laboratory of Millimeter Waves, School of Information Science and Engineering, Southeast University, Nanjing 210096, China; cjsu1990@163.com (C.S.); yqun.liu@gmail.com (Y.L.); 101010740@seu.edu.cn (H.Z.)

[2]  College of Electronic and Optical Engineering & College of Microelectronics, Nanjing University of Posts and Telecommunications, Nanjing 210003, China; liull@njupt.edu.cn

[3]  College of Telecommunication & Information Engineering, Nanjing University of Posts and Telecommunications, Nanjing 210003, China; yangm1010@njupt.edu.cn

*  Correspondence: 101010074@seu.edu.cn; Tel.: +86-139-1294-7543

**Abstract:** An experimental evaluation of multipath mitigation using a beam steering broadband circular polarization antenna (BSBCPA) in indoor passive localization system based on time differences of arrival (TDOA) is presented in this paper. The BSBCPA consists of a beam switch network, four identical hexagon patch elements and their respective feeding networks. By controlling the states of a radio frequency (RF) switch in the beam switch network, four steering circular polarization beams can be excited separately for azimuth omnidirectional coverage. Combining the spatial selectivity of steering beams and circular polarization in the BSBCPA, the positioning inaccuracy from indoor multipath propagation can be mitigated. In two different indoor environments with line of sight (LOS), complex multipath, when transmitting a 20 MHz bandwidth signal in WLAN, the 2D positioning mean error obtained is 0.7 m and 0.82 m, respectively. Compared with conventional omnidirectional linear polarization antenna (OLPA), the BSBCPA can at least improve positioning accuracy by 51%. The experimental results show that the proposed BSBCPA can significantly mitigate multipath propagation for TDOA-based indoor passive localization.

**Keywords:** TDOA; indoor passive localization; multipath mitigation; circular polarization; beam steering antenna

---

## 1. Introduction

Since the Global Positioning System (GPS) is not suitable for indoor applications, indoor localization has recently received more and more attention, especially in the areas of public safety, asset management, shopping and exhibition services [1,2]. Existing indoor localization methods, according to different technical categories, can be divided into acoustic, optical and RF methods.

Since RF signals can penetrate obstacles for localization and communication, indoor localization based on RF becomes the best candidate. The main metrics for estimating the target location using the RF method include: Received Signal Strength (RSS), Angle of Arrival (AOA), Time of Arrival (TOA) and TDOA [3]. Among these methods, the time-based positioning method (TOA or TDOA) is studied in the paper because of its high accuracy [4]. In particular, the TDOA method is more attractive as it

does not require information about the transmitting signal, which is also known as passive localization. Cross-correlation is widely adopted in TDOA-based localization system for time delay estimation [5]. Due to multipath effect in the indoor environment, however, the accuracy of estimated TDOA based on cross-correlation is severely degraded, especially in narrow-band spread spectrum system such as IEEE 802.11b WLAN.

In recent years, several methods have been proposed by researchers to mitigate the effect of multipath propagation on the accuracy of TDOA-based indoor localization. Super-resolution methods such as minimum-norm [6], root multiple signal classification (MUSIC) [7] and total least square-estimation of signal parameters via rotational invariance techniques (TLS-ESPRIT) [8] are well-known methods based on signal processing to achieve accurate multipath time delay estimation. As reported in [9], the positioning accuracy of 1–5 m can be realized using super-resolution methods. Nevertheless, in addition to the high computational complexity, these super-resolution methods can only be used in Orthogonal Frequency Division Multiplexing (OFDM) systems, and the number of multipath components needs to be estimated in advance. In [10], Exel et al. proposed a special synchronized transceiver for IEEE 802.11b, which can achieve timestamps with accuracy in the sub-nanosecond range. When tested in a conference room, the maximal possible accuracy of this positioning system based on timestamps is about 5 cm [11]. However, the timestamp method depends on the current physical and MAC layer of IEEE 802.11, which is difficult to apply to other modulation and data formats. A CLEAN deconvolution algorithm with a parameter optimization step is proposed in [12], which provides robustness for multipath mitigation. However, a specially designed impulsive symbol is needed to achieve good range estimation. As is known to all, multipath effect is not only related to the environment, but also to the antenna, especially to the radiation beamwidth and polarization of an antenna. Therefore, how to use the antenna to improve the positioning accuracy is a topic worthy of further exploration, and it is expected to become a widely applied method.

An experimental evaluation of mitigating multipath propagation based on antenna for TDOA-based indoor passive localization is proposed in this paper. With a combination of steering beams and circular polarization, the BSBCPA is designed and fabricated. Meanwhile, a TDOA-based receiver was designed to implement indoor location estimation. Unlike common RSSI-based indoor localization system using directional circular polarized antennas [13], the experiment in the paper is to use the BSBCPA to evaluate TDOA-based indoor localization without prior knowledge of the target direction. In two different indoor environments, the 2D positioning mean error is 0.7 m and 0.82 m with a 20 MHz bandwidth signal in WLAN and the accuracy is at least improved by 51% compared with the OLPA. To the best of the authors' knowledge, this is the first time the BSBCPA is used to evaluate multipath mitigation in TDOA-based indoor positioning system and ultimately achieve sub-meter positioning accuracy. Compared with other multipath mitigation methods, the proposed BSBCPA does not require complex calculation and can work independently of the modulation formats. In addition, it can be combined with the existing multipath mitigation methods based on signal processing to improve indoor positioning accuracy further.

The rest of this paper is organized as follows. Section 2 describes the principle of TDOA-based localization and indoor multipath analysis based on antenna characteristics. The specific localization system implementation is presented in Section 3, including antenna design, localization system architecture and algorithm implementation. The experimental results are given in Section 4. Finally, Section 5 concludes the paper.

## 2. Localization Principle and Multipath Analysis

### 2.1. Principle of TDOA-Based Localization

A TDOA-based localization system is used to determine the position of the transmitter by measuring the TDOA of the transmitted signal at pairs of receivers whose position is known, where the transmitter and receiver are also known as target node (TN) and reference nodes (RNs). The basic

structure of the system is shown in Figure 1, which includes several RNs and one TN. $d_i$ denotes Euclidean distance between the TN and the RN *i*:

$$d_i = \sqrt{(x - x_i)^2 + (y - y_i)^2} i, j = 1, 2, 3 \ldots \quad (1)$$

where $(x, y)$ and $(x_i, y_i)$ are the coordinate of the TN and the RN *i*. The distance difference between two RNs can be obtained by one TDOA estimation:

$$d_{ij} = d_i - d_j = c \cdot (\tau_i - \tau_j) = c \cdot \tau_{ij} \quad (2)$$

where $\tau_i$ denotes the propagation time of the signal from the TN to the RN *i*, *c* is the propagation speed of electromagnetic waves in the air. Therefore, the potential TN position is on a hyperbola with the focuses on the positions of the two RNs, and TDOA-based localization system is also known as hyperbolic localization system. For a 2D scenario, at least three RNs are required to locate one TN in TDOA-based localization system. Eventually, the TN position is determined from the intersection of multiple hyperbolas. Thus it can be seen that accurate TDOA estimation is the prerequisite of high-precision localization in the entire system.

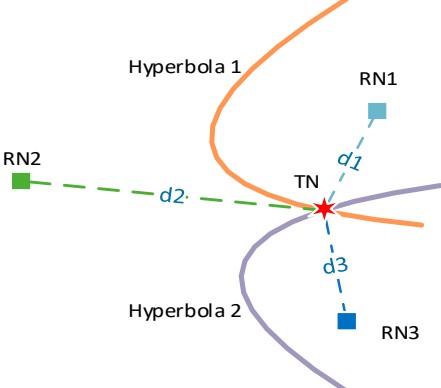

**Figure 1.** Basic architecture of TDOA-based localization system.

Cross-correlation method is a well-known technique for measuring time delay estimation and is used here for TDOA estimation. The signal received by two spatially separated RN *i* and *j* can be expressed as:

$$r_i(t) = s(t + \tau_i) + w_i(t) \quad (3)$$

$$r_j(t) = s(t + \tau_j) + w_j(t) \quad (4)$$

where $s(t)$ is transmitted signal by TN, $w(t)$ is the AWGN with zero mean. Then TDOA can be estimated by detecting the peak position of the cross-correlation function of the received signal $r_i(t)$ and $r_j(t)$:

$$C(\tau) = \int_0^T r_i(t) r_j(t + \tau)\, dt \quad (5)$$

$$\hat{\tau}_{ij} = \underset{\tau}{argmax}[C(\tau)] \quad (6)$$

where *T* is the observation time window.

### 2.2. Indoor Multipath Analysis Based on Antenna Characteristics

Typical indoor multipath propagation scenario is shown in Figure 2. In addition to a direct path signal, reflected signals propagating along non-direct paths are also present in the indoor wireless channel. The impulse response of an indoor wireless channel in time domain can be expressed as

$$h(t) = \sum_{k=0}^{N} a_k \delta(t - \tau_k) \tag{7}$$

where $N$ is the number of multipath components, $a_k = |a_k|e^{j\theta_k}$ is the complex attenuation, $\tau_k$ is the $k$th multipath propagation delay. Due to very small changes in parameters $a_k$ and $\tau_k$ within one snapshot of localization measurement, the indoor channel can be treated as time-invariant channel. Therefore, the signal received by the antenna is actually an overlap of the signals with different delays propagating along multipath, and only the signal propagating along the direct path can be used for localization. When TDOA is estimated in a dense indoor multipath scenario, multiple correlation peaks generated by signals containing different delays can be overlapped after cross-correlation. This mainly occurs in narrowband signals with wider correlation peaks [14], such as WLAN signals. The overlap of multiple peaks has a wider peak, resulting in an erroneous TDOA estimation. As the propagation speed of electromagnetic waves is constant $c = 3 \times 10^8$ m/s, even a very small delay estimation error, such as a few ns, will cause a meter-level distance error. Because of the limitation of sample rate and signal bandwidth, it is difficult to distinguish direct path signals from received signals in time domain. As we all know, multipath effect is not only related to the inherent characteristics of the channel, but also related to the beamwidth and polarization of the receiving antenna.

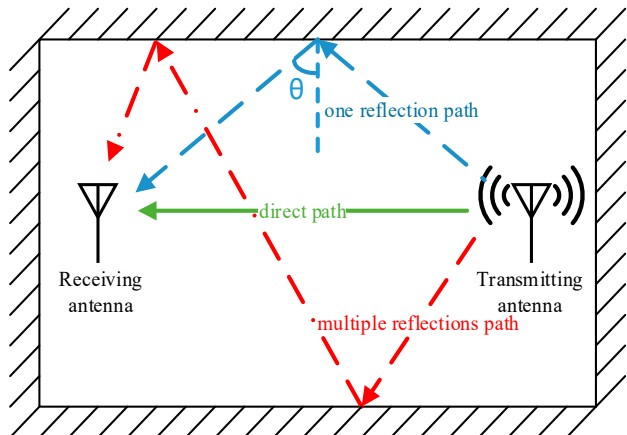

**Figure 2.** Indoor multipath propagation scenario.

The receiving antenna beamwidth is directly related to the number of multipath components. Therefore, in many literatures, we can see localization measurements with directional antennas instead of omnidirectional antennas. Generally, the narrower the antenna beamwidth, the better the multipath propagation mitigation. But, it is not easy to realize a very narrow beamwidth unless massive antenna arrays are used. In addition, without knowing the direction of transmitting antenna, the narrow beam must be capable of omnidirectional scanning, either electronically or mechanically, which will undoubtedly increase the cost. By exploiting the spatial selectivity of a beam steering antenna, multipath components can be mitigated at low cost.

The circular polarization characteristic of the antenna can significantly mitigate multipath propagation. It is well known that when a circularly polarized wave is incident on a conductive reflecting surface, the polarization state of one-order reflective wave changes, while it depends on the incident angle and dielectric constant of the reflective material when the reflective material is non-conductor. For typical non-conductive reflective materials in the indoor environment, such as concrete, wood and glass, when the incident angle is small, the polarization direction of one-order reflected wave is opposite to that of the incident wave. Therefore, when the receiving antenna polarization is the same as the transmitting antenna, part of one-order reflected wave can be attenuated by cross-polarization isolation. Due to the large propagation loss and reflection loss, two or more reflections have less influence on the positioning accuracy.

## 3. Localization System Implementation

### 3.1. Antenna Design

Figure 3a,b, respectively, depicts the configuration of the proposed BSBCPA and its feeding network. The BSBCPA consists of four identical hexagon patch elements and their respective feeding networks and a beam switch network placed below the patch elements. The hexagon patch element is fed by three capacitive disks in the same layer to compensate for the inductance caused by long probes. The feeding capacitive disks are directly connected to the output ports of the feeding network via three long metallic probes passing through an air gap of height H. In order to achieve broadband circular polarization, equal amplitudes and relative 120° phase differences are required between three feeding ports [15]. Therefore, a Wilkinson power divider with three output ports is introduced as the feed network, and there is a one-third wavelength difference between the output ports, where port 1 is the input port and ports 2, 3, and 4 are the output ports. In addition, for good AR bandwidth, low coupling between output ports of the feeding network should be guaranteed. Therefore, half-wavelength transmission lines are introduced to bridge the isolation resistances between the output ports of the power divider, as they do not change the impedance characteristics of the transmission line. In order to match the output ports, the added isolation resistances should be 150 ohms. The feeding networks are connected to the beam switch network via cables.

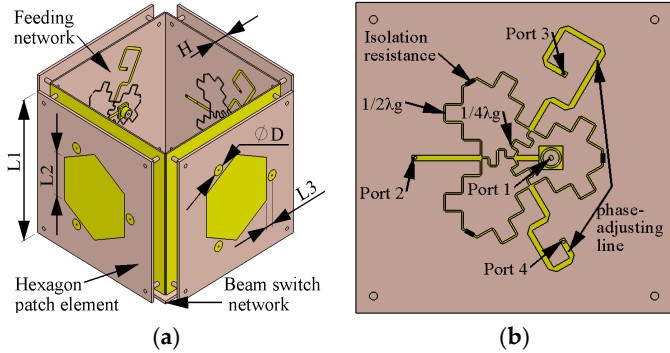

**Figure 3.** (**a**) Structure of the proposed BSBCPA. (**b**) Structure of the feeding network (λg is the wavelength in the substrate of the feeding network).

A single-pole four-throw RF switch PE42641 is introduced in the beam switch network. By controlling the RF switch, the beam switch network can offer four states, and each state is corresponding to a steering circular polarization radiation pattern. Furthermore, since low signal-to-noise ratio (SNR) can significantly reduce the accuracy of TDOA estimation, a low noise amplifier (LNA) BGB741LESD is also used in the beam switch network to keep a low noise figure of the receiver. The center frequency of the antenna is chosen at 2.45 GHz. The hexagon patch element and feeding network are developed on FR4 substrates (dielectric constant 4.4 and loss tangent 0.025) with heights of 1.6 mm and 0.8 mm. The detailed geometrical parameters of the proposed antenna are shown in Table 1.

**Table 1.** Geometrical parameters.

| Parameters | Value (mm) | Parameters | Value (mm) |
|:---:|:---:|:---:|:---:|
| L1 | 85.5 | D | 5.6 |
| L2 | 28 | H | 8 |
| L3 | 3.8 | λg | 72 |

A prototype of the proposed antenna is fabricated and measured for reflection coefficient and radiation pattern. The reflection coefficient measurements were carried out by the Keysight Technologies vector network analyzer E5071C. The antenna radiation pattern and AR was measured

using a rotating linear polarization transmitting horn as an auxiliary antenna in an anechoic chamber. The measured reflection coefficient and AR are shown in Figure 4 along with the simulated results. The measured 10 dB impedance bandwidth and 3 dB AR bandwidth are from 2 to 3 GHz and from 2 to 2.8 GHz, respectively. The overall trends of simulation and measurement fit well with each other. The slight differences between them are mainly due to the uncertain permittivity of the substrate and fabricating errors. Figure 5 shows the measured radiation patterns of four states in the azimuth plane at 2.45 GHz. The ripples in the radiation pattern are a consequence of the beam ellipticity, which occurs when a finite cross-polarization component exists [16]. The larger the ripples in the pattern, the larger the cross-polarization levels. As shown in radiation pattern of every state, the AR is almost less than 3 dB within a 90° beamwidth. These measured results confirm that the BSBCPA achieves azimuth omnidirectional coverage.

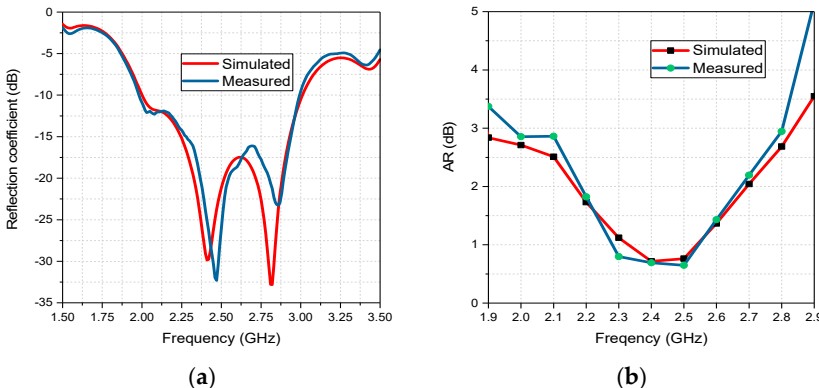

(a)　　　　　　　　　　　　　　　　　　　　(b)

**Figure 4.** (**a**) Simulated and measured reflection coefficients. (**b**) Simulated and measured ARs.

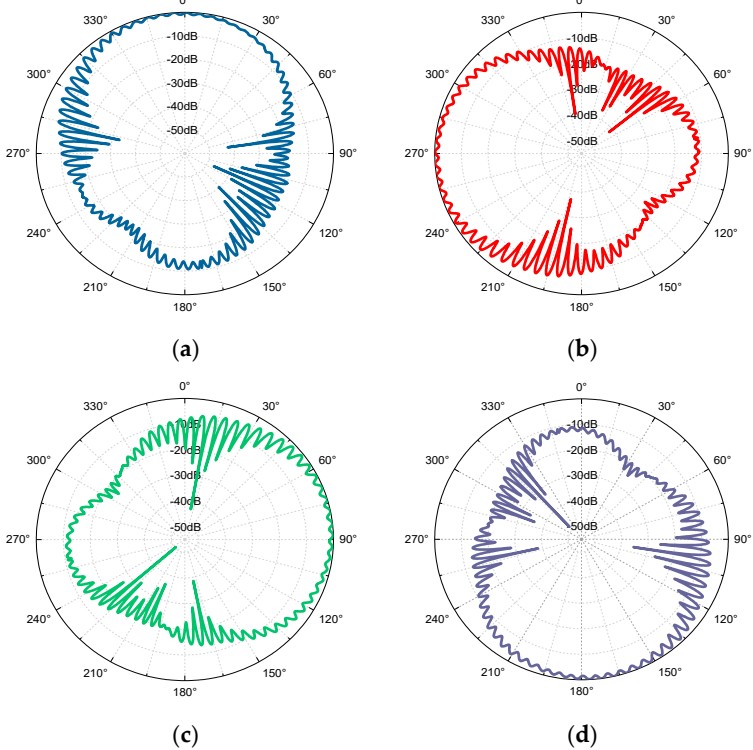

**Figure 5.** Measured radiation patterns of four states in the azimuth plane. (**a**) Radiation at 0°. (**b**) Radiation at 270°. (**c**) Radiation at 90°. (**d**) Radiation at 180°.

### 3.2. Localization System Architecture

The implemented TDOA-based 2D indoor localization system is composed of three RNs and one TN. The TN includes a transmitter and an omnidirectional broadband circularly polarization transmitting antenna. By introducing the curved branches at both the edges of the patch and ground plane in the transmitting antenna, the omnidirectional circular polarization fields can be obtained in the wider azimuth planes [17]. The R&S vector signal generator SMBV100A is applied as the transmitter to generate BPSK modulation signal with a bandwidth of 20 MHz in WLAN. Each RN consists of the BSBCPA as receiving antenna, a receiver and a PC that records and transmits data to a workstation for offline processing.

The hardware architecture of the receiver is shown in Figure 6. Behind receiving antenna, a two-stage LNA and a RF filter are respectively used to amplify the received signal and suppress out of band interference. Then the received RF signal is demodulated to the baseband signal consisting of the in-phase and quadrature component by a local oscillator (LO). Subsequently, the baseband signal is filtered by a low-pass filter with a 3 dB cut-off frequency of 25 MHz and amplified by a variable gain amplifier to maintain the output voltage within an appropriate range. After 80 MSps ADC sampling, when a trigger signal is received, the digital signal is temporarily stored in the buffer memory by the FPGA and finally uploaded to the PC.

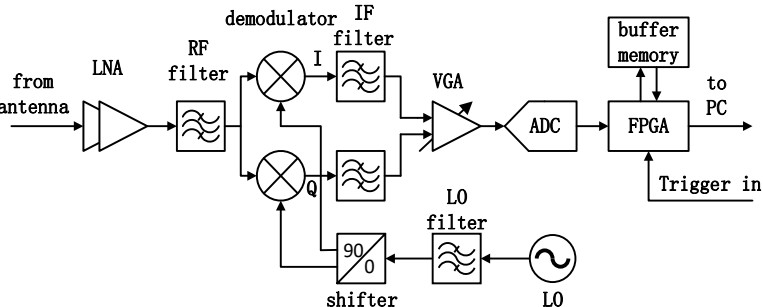

**Figure 6.** Hardware architecture of the receiver.

In TDOA-based localization system, strict synchronization must be performed between the RNs, including time and frequency synchronization. For outdoor localization, a GPS disciplined oscillator (GPSDO) is typically used to synchronize the distributed RNs. Considering that it is difficult to receive GPS signal indoors, and the time standard deviation of existing GPSDO is pretty large for indoor localization (usually about 50 ns) [18], a common reference clock module is used to synchronize all RNs through cables. This module includes a temperature compensated crystal oscillator providing a stable 10 MHz reference clock and a microcontroller unit generating trigger signal, which are used to realize frequency and time synchronization between the RNs, respectively.

### 3.3. Algorithm Implementation

Since the data used for the cross-correlation is a sampled discrete signal, the time resolution of the TDOA estimation is severely limited by the sample rate of the receiver. For example, if the sample rate $f_s$ used in this receiver is 80 MSps, the time interval between two successive samples is $T_s = 1/f_s = 12.5$ ns and the corresponding distance resolution is 3.75 m. In order to further improve the time resolution of the TDOA measurement, a quadratic interpolation [19] is adopted as an optimized time delay estimation algorithm to achieve sub-sample level accuracy in the indoor localization system. Generally, the estimated cross-correlation function has a close shape to a convex parabola in the neighborhood of its maximum. Therefore, the cross-correlation function in the peak position can be modeled as

$$C(\tau) = a\tau^2 + b\tau + c \tag{8}$$

The unknown constant in (1) can be uniquely determined by the peak from the cross-correlation function and its two adjacent points. Then the TDOA can be estimated by the apex of the parabola:

$$\hat{\tau}_{ij} = -\frac{b}{2a} \tag{9}$$

In the Workstation, in addition to the TDOA estimation, position calculation and display, the outlier elimination is also fulfilled in NI LabVIEW development environment. Due to the influence of wireless channel noise, outliers will appear in TDOA estimation and cause large positioning errors. They are typically only a small proportion of measurements and have a small correlation coefficient. During every TDOA estimate, estimates that differ from the estimated value with the largest correlation coefficient by more than one sampling interval are considered to be outliers and are automatically removed.

## 4. Result and Discussion

Two experiments were carried out in a real environment. In the first experiment, RF cables were used as channels to avoid the multipath propagation. In the second experiment, two different indoor environments, the conference room and the laboratory, were used for 2D indoor location estimation. Both places are static environments with LOS and complex multipath, and the measurement scenario in the conference room is shown in Figure 7. By adjusting the output power of the transmitter, at least SNR of 10 dB is guaranteed in every TDOA experiment. Each position in the statistical result is estimated by 100 TDOA calculations, and 1024 samples are applied for each TDOA calculation.

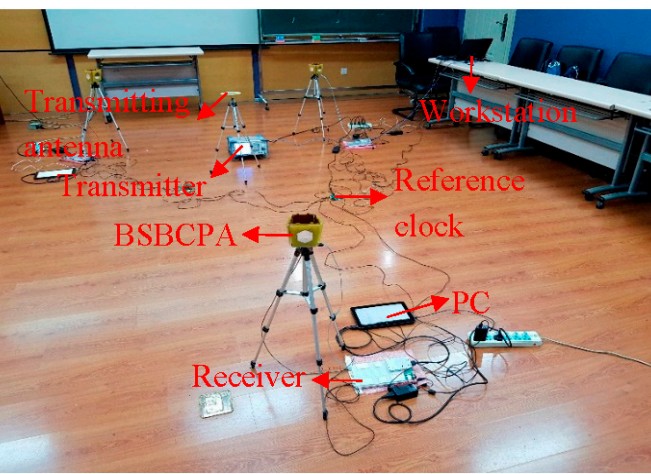

**Figure 7.** Photo of positioning equipment and venue in conference room.

*4.1. System Evaluation Using RF Cables*

This experiment was carried out to assess the performance of the implemented receiver and algorithm. Several cables of known length and a power divider were used in this experiment. As long as the propagation speed of electromagnetic wave in the cable is known (0.7 ×c was used in the experiment), the length difference between cables can be estimated from the obtained TDOA.

Due to differences among individual devices, there may be ns-level delay difference for devices of the same model, and there will still be delay difference even if the same device is in different states (such as different temperature or gain). Therefore, two equal length cables were used firstly to calibrate the inherent time difference between two receivers.

Table 2 summarizes the statistical results for all different length cables. It can be seen that the maximum of all the mean error of estimated length difference between cables is 7 cm, which proves the performance of the implemented receiver and algorithm without multipath effect.

**Table 2.** Statistical results for different length cables.

| Length Difference between Cables (m) | Mean Error (cm) | Standard Variance |
|:---:|:---:|:---:|
| 1 | 6 | 0.01 |
| 2 | 4 | 0.02 |
| 3 | 7 | 0.02 |

*4.2. 2D Indoor Location Estimation*

In this experiment, the BSBCPA and OLPA were used separately as receiving antenna to evaluate the multipath propagation mitigation. Both the transmitting and receiving antennas were placed on tripods at an altitude of 0.8 m from the ground. For convenience, the transmitter (TX) and the two receivers (RX1, RX2) are fixed, and the third receiver (RX3) is moved around in steps of 1 m for a total of 20 positions. The specific measurement configuration is shown in Figure 8. When the BSBCPA was used, omnidirectional scanning was first performed by switching four beams of the BSBCPA after each shift of RX3 without prior knowledge of the target direction, and the received signal in each beam was uploaded to the PC for comparison. Since the received power is maximum when there is a direct path between the transmitting and the receiving antenna, the excited patch element for localization is selected by the received signal having the maximum power. TDOA-based positioning measurements were performed after determining the excited patch element.

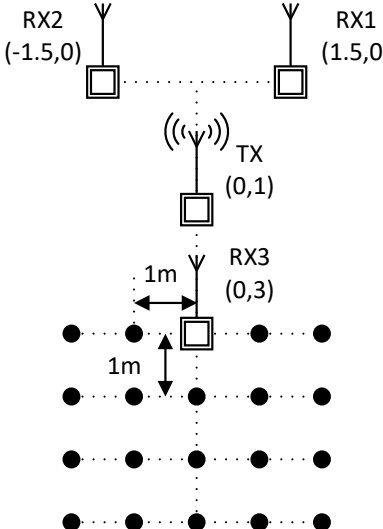

**Figure 8.** Measurement configuration of 2D location estimation.

When the RX3 is at location (0, 6), the positioning results for one time and 100 times are shown in Figure 9. The positioning results for each location in two different indoor environments are shown in Table 3. It can be seen that in most positions, the BSBCPA performs better. The mean error at all positions using the BSBCPA and OLPA in the conference room is 0.7 m and 1.53 m respectively, and the results in the laboratory is 0.82 m and 1.7 m, indicating that the positioning accuracy improved by at least 51%. The cumulative distribution function (CDF) of the positioning error is shown in Figure 10. Similarly, the BSBCPA performs better with a median error of 0.62 m in the conference room, and the OLPA only achieved a median error of 1.39 m. In the laboratory, the BSBCPA and OLPA achieved median errors of 0.76 m and 1.27 m, respectively.

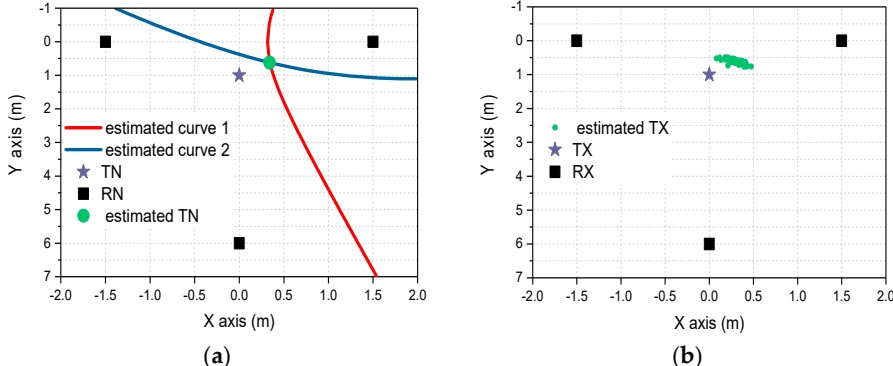

**Figure 9.** (**a**) Position estimation of one TDOA calculation. (**b**) Position estimation of 100 TDOA calculations.

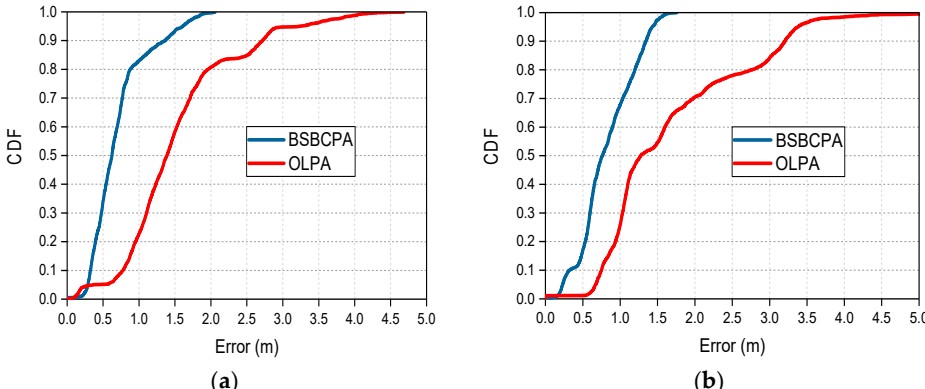

**Figure 10.** (**a**) CDF of the positioning error in the conference room. (**b**) CDF of the positioning error in the laboratory.

**Table 3.** Statistical results of 2D indoor location estimation.

| Position of RX3 | Conference Room | | | | Laboratory | | | |
|---|---|---|---|---|---|---|---|---|
| | BSBCPA | | OLPA | | BSBCPA | | OLPA | |
| | Mean Error (m) | Standard Variance | Mean Error (m) | Standard Variance | Mean Error (m) | Standard Variance | Mean Error (m) | Standard Variance |
| (0, 3) | 0.54 | 0.07 | 1.69 | 0.16 | 0.25 | 0.09 | 1.55 | 0.21 |
| (1, 3) | 0.81 | 0.07 | 1.33 | 0.10 | 0.67 | 0.07 | 0.80 | 0.09 |
| (2, 3) | 0.58 | 0.12 | 1.43 | 0.07 | 0.45 | 0.06 | 0.97 | 0.07 |
| (−1, 3) | 0.73 | 0.10 | 1.76 | 0.20 | 1.08 | 0.14 | 2.90 | 0.62 |
| (−2, 3) | 1.65 | 0.25 | 1.82 | 0.18 | 1.16 | 0.20 | 2.12 | 0.48 |
| (0, 4) | 0.37 | 0.06 | 1.77 | 0.16 | 1.03 | 0.26 | 2.09 | 0.15 |
| (1, 4) | 0.75 | 0.10 | 0.84 | 0.08 | 0.58 | 0.12 | 0.99 | 0.11 |
| (2, 4) | 0.57 | 0.10 | 1.11 | 0.07 | 1.16 | 0.10 | 1.00 | 0.09 |
| (−1, 4) | 0.36 | 0.06 | 0.90 | 0.06 | 0.59 | 0.08 | 1.05 | 0.05 |
| (−2, 4) | 1.52 | 0.16 | 3.73 | 0.33 | 1.23 | 0.18 | 3.10 | 0.25 |
| (0, 5) | 0.63 | 0.03 | 1.08 | 0.07 | 0.82 | 0.12 | 1.13 | 0.09 |
| (1, 5) | 0.40 | 0.23 | 1.20 | 0.09 | 0.74 | 0.21 | 1.53 | 0.06 |
| (2, 5) | 0.73 | 0.05 | 2.59 | 0.13 | 0.59 | 0.05 | 3.21 | 0.11 |
| (−1, 5) | 1.17 | 0.23 | 1.34 | 0.13 | 0.92 | 0.17 | 1.10 | 0.10 |
| (−2, 5) | 0.72 | 0.12 | 1.47 | 0.18 | 0.95 | 0.13 | 3.28 | 0.33 |
| (0, 6) | 0.49 | 0.03 | 0.18 | 0.07 | 0.59 | 0.05 | 0.65 | 0.06 |
| (1, 6) | 0.34 | 0.06 | 2.75 | 0.10 | 1.34 | 0.05 | 1.64 | 0.09 |
| (2, 6) | 0.46 | 0.06 | 1.02 | 0.10 | 0.23 | 0.04 | 0.74 | 0.06 |
| (−1, 6) | 0.30 | 0.05 | 1.98 | 0.52 | 0.59 | 0.05 | 3.13 | 1.34 |
| (−2, 6) | 0.98 | 0.13 | 0.70 | 0.08 | 1.43 | 0.12 | 1.15 | 0.13 |
| Mean | 0.70 | - | 1.53 | - | 0.82 | - | 1.70 | - |

## 5. Conclusions

This paper presents an experimental evaluation of mitigating multipath propagation using a BSBCPA in TDOA-based indoor passive localization system. In two different indoor environments with LOS, complex multipath, compared with conventional OLPA, the localization system with the BSBCPA can improve positioning accuracy by at least 51%. According to the positioning results, the proposed BSBCPA can significantly mitigate multipath propagation and improve positioning accuracy for TDOA-based indoor localization, and is expected to promote the development of indoor localization in the future.

**Author Contributions:** Conceptualization, X.Y.; Data curation, L.L. and M.Y.; Methodology, H.Z.; Software, C.S. and Y.L.; Validation, C.S.; Writing—original draft, C.S. and X.Y.

**Funding:** This research was funded by National Natural Science Foundation of China, grant number 61427801, 61771127, U1536123 and U1536124.

**Conflicts of Interest:** The authors declare no conflict of interest.

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
