# Peer review of "Experimental Evaluation of Multipath Mitigation in TDOA-Based Indoor Passive Localization System Using A Beam Steering Broadband Circular Polarization Antenna"

_electronics, doi:10.3390/electronics7120362_

Round 1
Reviewer 1 Report
The manuscript describes an evaluation of mitigating multipath propagation using a beam steering broadband 14 circular polarization antenna in TDOA-based indoor localization system.
The performance of the system is evaluated by real experiments ielding an average error about 0.8 metres.
I think that it is correctly written and organized with relevant and novel information. Related work is well described and updated.
I find the work interesting and useful. My recommendation is to accept after minor revisions.
Some comments:
Lines 38-40. Authors said "Among these methods, the time-based positioning method (TOA or TDOA) is the most commonly used and studied because of its higher accuracy and smaller system complexity [4]." TOA or TDOA systems have high accuracy, however, indoor localization systems based on figerprinting of RSS are the most used systems. Furthermore, in TOA and TDOA systems additional and specific hardware is needed, such as is decribed in this mansucript (antenna) and strict synchronization must be performed. Please, re-write the sentence.
At the end of Section 1, a paragraph should be introduced describing the structure of the manuscript.
Line 80. 3.1 --> 2.1
Line 111. 3.2 -- 2.2
Author Response
Point 1: Lines 38-40. Authors said "Among these methods, the time-based positioning method (TOA or TDOA) is the most commonly used and studied because of its higher accuracy and smaller system complexity [4]." TOA or TDOA systems have high accuracy, however, indoor localization systems based on fingerprinting of RSS are the most used systems. Furthermore, in TOA and TDOA systems additional and specific hardware is needed, such as is described in this manuscript (antenna) and strict synchronization must be performed. Please, re-write the sentence.
Response 1: The authors would like to acknowledge and express their appreciation to the reviewer for the valuable suggestions and helping the authors improve the quality of this manuscript. The authors agree with the viewpoint revealed in the point 1 that the most widely used indoor localization system is based on fingerprinting of RSS and the time-based localization has large system complexity.
The sentence "Among these methods, the time-based positioning method (TOA or TDOA) is the most commonly used and studied because of its higher accuracy and smaller system complexity [4]" has been corrected to "Among these methods, the time-based positioning method (TOA or TDOA) is studied in the paper because of its high accuracy".
Point 2: At the end of Section 1, a paragraph should be introduced describing the structure of the manuscript.
Response 2: The manuscript has been revised according to this comment. A paragraph has been introduced describing the structure of the manuscript at the end of Section 1, as follows.
The rest of this paper is organized as follows. Section 2 describes the principle of TDOA-based localization and indoor multipath analysis based on antenna characteristics. The specific localization system implementation is presented in Sections 3, including antenna design, localization system architecture and algorithm implementation. The experimental results are given in Sections 4. Finally, Section 5 concludes the paper.
Point 3: Line 80. 3.1 --> 2.1 Line 111. 3.2 -- 2.2

Response 3: The manuscript has been revised according to this comment.
The error "3.1" has been corrected to "2.1". The error "3.2" has been corrected to "2.2".
Reviewer 2 Report
It is a very interesting work with validated experimental results. However, some syntax issues should be corrected. For example, in the Introduction it is written that:
"Because RF signal can penetrate obstacles and can be used simultaneously for localization and communication"
Author Response
Point 1: It is a very interesting work with validated experimental results. However, some syntax issues should be corrected. For example, in the Introduction it is written that:
"Because RF signal can penetrate obstacles and can be used simultaneously for localization and communication" 

Response 1: The authors would like to acknowledge and express their appreciation to the reviewer for the valuable suggestions about some syntax issues.
The sentence "Because RF signal can penetrate obstacles and can be used simultaneously for localization and communication" has been corrected to "Since RF signals can penetrate obstacles for localization and communication".